# Designing efficient genetic code expansion in Bacillus subtilis to gain biological insights

Devon A. Stork[1,2], Georgia R. Squyres [2], Erkin Kuru[1,3], Katarzyna A. Gromek[4], Jonathan Rittichier[1,3], Aditya Jog[1], Briana M. Burton [4], George M. Church[1,3 ✉], Ethan C. Garner [2 ✉] & Aditya M. Kunjapur [1,5 ✉]

*Bacillus subtilis* is a model gram-positive bacterium, commonly used to explore questions across bacterial cell biology and for industrial uses. To enable greater understanding and control of proteins in *B. subtilis*, here we report broad and efficient genetic code expansion in *B. subtilis* by incorporating 20 distinct non-standard amino acids within proteins using 3 different families of genetic code expansion systems and two choices of codons. We use these systems to achieve click-labelling, photo-crosslinking, and translational titration. These tools allow us to demonstrate differences between *E. coli* and *B. subtilis* stop codon suppression, validate a predicted protein-protein binding interface, and begin to interrogate properties underlying bacterial cytokinesis by precisely modulating cell division dynamics in vivo. We expect that the establishment of this simple and easily accessible chemical biology system in *B. subtilis* will help uncover an abundance of biological insights and aid genetic code expansion in other organisms.

[1] Department of Genetics, Harvard Medical School, Boston, MA, USA. [2] Department of Molecular and Cellular Biology, Harvard University, Cambridge, MA, USA. [3] Wyss Institute for Biologically Inspired Engineering, Boston, MA, USA. [4] Department of Bacteriology, University of Wisconsin-Madison, Madison, WI, USA. [5] Present address: Department of Chemical and Biological Engineering, University of Delaware, Newark, DE, USA. ✉email: gchurch@genetics.med.harvard.edu; egarner@g.harvard.edu; kunjapur@udel.edu

**B**acillus subtilis is a gram-positive soil bacterium naturally found in the plant rhizosphere[1,2] and employed in basic and applied research[3]. Genetic tractability and ease of culture have made B. subtilis the most commonly used gram-positive model organism. In fundamental research, B. subtilis is a model organism for the study of endospore formation[4], asymmetric cell division[5], biofilm formation[6], and multicellular behavior[7]. In applied research, B. subtilis is a probiotic for plants and animals, including humans, spanning agricultural, nutritional, and medical applications[8–10] and is used for industrial protein production[11]. A broad range of available genetic tools, including inducible promoters and protein tags[5,12] have made B. subtilis an attractive model for both fundamental research and industrial biotech, but limitations in the chemistry and flexibility of these tools prevent full utilization of the organism's potential. The recent use of CRISPRi[13] and optogenetic transcriptional control[14] transfer tools from E. coli to B. subtilis, though each is limited in their expression range and titratability. New types of peptidoglycan tags[15] allow chemical modification and imaging of the development of the cell wall, but not proteins. To examine broader mechanisms of growth and division in B. subtilis, finer chemical functionalization and titration tools are needed. We postulate that genetic code expansion through the incorporation of nonstandard amino acids (nsAAs) in proteins could achieve that in B. subtilis as has been demonstrated in E. coli.

The technique of site-specific nsAA incorporation widens the chemistry accessible to biological systems by allowing the use of diverse chemical functional groups in protein design and experimentation[16,17]. More than 200 different nsAAs derived from tyrosine, pyrrolysine, serine, leucine, and tryptophan have been incorporated into proteins, primarily in E. coli and mammalian systems[18]. Engineered variants of the corresponding orthogonal amino-acyl-tRNA synthetase (AARS) and tRNA pairs are used to attach nsAAs to tRNA that direct them for ribosomal incorporation at specific codons. An engineered AARS/tRNA system must be orthogonal to the native translational machinery of the host organism; thus, separately engineered systems must often be used in different organisms[16]. The requirements of orthogonality and balanced expression levels of AARS and tRNA have limited the utility of genetic code expansion in bacterial systems outside of E. coli[19,20].

Here we provide a broad investigation of the portability of many different sets of genetic code expansion technologies from E. coli to B. subtilis, and we use several of these systems to gain insights about protein translation and cell division in B. subtilis. This work builds substantially upon recent demonstrations of genetic code expansion in B. subtilis, where single pyrrolysine[21] and tyrosine[22] nsAAs were incorporated. We expand the number of nsAAs in B. subtilis to 20, using 3 families of stable, genomically integrated AARS constructs and achieve higher incorporation efficiency than in previous studies. The functions of these nsAAs range from bio-orthogonal tagging to photocrosslinking and to fluorescence, with broad experimental utility. To facilitate the further application of this technology and explain complications noted in previous work[21,22], we firstly show that unlike in nonrecoded E. coli[23–25], nsAAs incorporate efficiently at amber stop codons in native B. subtilis genes. Second, the incorporation of photocrosslinking nsAAs allows the demonstration of binding interactions of secreted proteins homologous to virulence factors. While previous work has shown nsAAs can modulate translational rates[22], we demonstrate a tighter system with a larger dynamic range and make a detailed comparison of nsAA titration to many different B. subtilis promoters. Finally, we use this system to facilitate biological discovery. We are able to precisely modulate the dynamics of the division protein FtsZ, an essential filament responsible for coordinating cell division[26]. Our data support the theory that FtsZ filaments must be above a minimal length to accomplish cell division[27]. These results and our deposited strains (at the Bacillus Genomic Stock Center) will facilitate the use of nsAAs for general use in B. subtilis across research and industrial applications.

## Results

### Activity of diverse orthogonal amino-acyl tRNA synthetases in B. subtilis.

To begin testing genetic code expansion in B. subtilis, we genomically integrated a codon-optimized Methanococcus jannaschii-tyrosyl-tRNA synthetase (MjTyrRS) variant called bipARS capable of incorporating the nsAA **1** at the lacA locus. This synthetase was accompanied by the corresponding tRNA and a panel of constitutive promoters driving AARS and tRNA expression. An IPTG-inducible reporter cassette was integrated at the amyE locus for the expression of an mNeongreen fluorescent protein with a TAG stop codon at position 2 (Fig. 1A). After an initial promoter screen with **1**, we determined that a pVeg/pSer AARS/tRNA promoter combination yielded the best incorporation with low background (Supplementary text & Supplementary Figure 1A–C), and **1** incorporation within mNeongreen was confirmed by mass spectrometry (Supplementary Figure 2E).

To explore the potential of enabling more diverse chemical functionality in B. subtilis, we sought to incorporate additional nsAAs that contain sidechains known to function as fluorescent probes, handles for click chemistry, and photo-crosslinkers. To this end, we built multiple additional AARS cassettes. First, we created more MjTyrRSs variant cassettes to diversify the tyrosine-based nsAAs available. Second, to investigate the use of nsAAs that are based on non-phenyl sidechains, we used the Saccharomyces cerevisiae tryptophan synthetase (ScWRS)[28], and the abkRS variant of the Methanosarcina barkeri pyrrolysine synthetase (MbPylRS)[29]. However, the MbPylRS was inactive in B. subtilis (Supplementary Figure 1D). A possible explanation for this lack of activity is the low solubility of the MbPylRS N-terminal domain in bacteria[30,31]. The homologous Methanomethylophilus alvus pyrrolysine synthetase (MaPylRS)[32] lacks this domain, and showed activity in Bacillus subtilis. Encouragingly, the incorporation activity of these diverse synthetases on their corresponding nsAAs (**1–6**) reflected published activity of corresponding synthetases in E. coli[18,33] (Fig. 1B–D), which suggests that AARS engineering performed in E. coli would allow predictable activity and specificity in B. subtilis. We utilized the substrate promiscuity of the MjTyrRSs to incorporate many more nsAAs, bringing the total incorporated in B. subtilis up to 20 (Supplementary Figure 1E, F), including nsAAs capable of fluorescence, photocrosslinking, click chemistry, metal chelation, and more. These nsAAs cover most of the applications of genetic code expansion.

To confirm nsAA incorporation, we expressed FLAG-tagged mNeongreen with nsAAs incorporated into an elastin-like peptide optimized for mass-spectrometry detection of nsAAs[34]. Purification and analysis of peptides demonstrated incorporation of all the tyrosine-based nsAAs shown in Fig. 1B (Supplementary Figure 2). Instead of detecting **5** at the indicated position, lysine was detected instead. This is likely due to the known deprotection of boc during the chromatographic step of peptide identification[35]. Additionally, **6** was not detected likely due to the inability to purify sufficient protein due to low-expression levels (Supplementary Figure 1G).

### Proteome-wide incorporation of nsAAs at TAG sites.

One potential application of nsAA incorporation is fluorescent tagging for subcellular microscopy[36,37]. We attempted to use nsAA **3** incorporation to specifically localize small, dynamic cell division

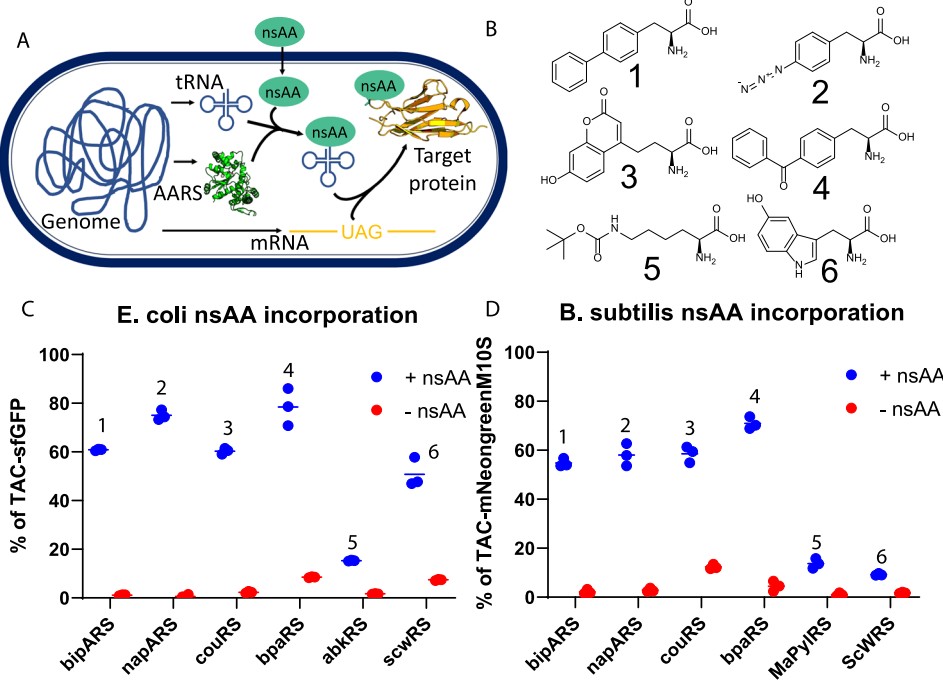

**Fig. 1 Genetic code expansion via nsAA incorporation in *B. subtilis*. A** Scheme of nsAA incorporation, with genomically integrated AARS and tRNA constructs incorporating externally provided nsAA into a genomically expressed gene containing an in-frame TAG amber stop codon. **B** Chemical structure of six nsAAs of primary interest to this study. **1**:biphenylalanine (bipA), **2**:p-azidophenylalanine (pAzF), **3**:Coumarin-nsAA (CouAA), **4**:Benzoylphenylalanine (BpA), **5**:boc-Lysine (boc-K), **6**:5-Hydroxytryptophan (5OHW) **C&D** nsAA incorporation from three biologically independent cultures in **C** *E. coli* and **D** *B, subtilis*. nsAA used is indicated above datapoints, synthetase variant below each *x*-axis. The MaPylRS is a homologous variant of abkRS, lacking an insoluble N-terminal domain. In all cases, signal is normalized to an identical reporter containing a TAC Tyr codon in place of a TAG amber codon. Individual datapoints for biological triplicates shown, with a horizontal bar at mean. In **C**, C321.ΔA recoded *E. coli* from plasmid-based AARS & tRNA and a genomic sfGFP reporter containing a single TAG codon in an N-terminal linker. **D** nsAA incorporation in *B. subtilis* from a genomic AARS & tRNA and a genomic M10SmNeongreen reporter containing a single TAG codon at position 2, immediately following the start codon.

components that have not previously been tagged. However, high background fluorescence hindered this goal (Supplementary Figure 3). The fluorescent coumarin nsAA was washed out of cells effectively in the absence of synthetase & cognate tRNA; however, synthetase & tRNA expression was sufficient for cellular retention of fluorescent nsAA (Supplementary Figure 4A). Whole protein lysate analysis revealed that the fluorescent nsAA was incorporated into many proteins across the proteome (Fig. 2A). This is a surprising phenomenon, for while sporadic incorporation of nsAAs into native proteins has been observed in wild-type *E. coli*[38], broad proteomic incorporation is an unknown phenomena[23–25]. Since the synthetase & tRNA are here expressed from single-copy genomic loci using native promoters, it is unlikely that they are significantly overexpressed, especially in comparison to plasmid-based synthetase systems used in *E. coli*[39]. Therefore, these results join other work[40] in suggesting that *B. subtilis* suppresses stop codons at a high rate and may be insensitive to Rho-dependent transcription termination of mRNAs containing premature stop codons[41].

To enrich for the native protein where nsAA is incorporated, nsAA **2** was incorporated for click-chemistry-based enrichment of proteins containing an nsAA. Mass-spectrometry analysis of enriched proteins revealed that nsAA incorporation followed by click-pulldown enriched for proteins ending with a TAG stop codon as compared to TAA or TGA (Fig. 2B). Analysis of the identity of the highly enriched proteins revealed that 23 of the 569 TAG-containing proteins made up 71% of the enriched events (Supplementary Table 1). Despite the apparent suppression of genomic TAG stop codons, no significant decrease in doubling times was observed (Supplementary Figure 4B). These results

explain and confirm speculation in previous works that nsAAs may incorporate into native genes in *B. subtilis*[21].

Since general incorporation into the *B. subtilis* proteome will interfere with specific labelling approaches, we sought to reduce the level of native protein incorporation. One approach to reduce the level of background nsAA incorporation and to further extend nsAAs in *B. subtilis* would be to use a codon rarer than the TAG codon in *B. subtilis*, such as the quadruplet TAGA codon. To do this, we cloned quadruplet versions of tyrosine synthetase cassettes. These contained a tRNA with a quadruplet anticodon and synthetases modified with the F261S and D286E mutations, which have been shown to encourage UCUA-tRNA aminoacylation for p-aceytlphenylalanine[39]. This quadruplet system was able to successfully incorporate at TAGA codons with low efficiency but also incorporated into the TAG codon with similar efficiency (Supplementary Figure 4C). This is not the only report of lack of specificity of the TAGA-tRNAs[42] and suggests future issues for attempts to use quadruplet codons to increase the available codon space. In addition to the lack of specificity for the TAGA, 217 of the 596 *B. subtilis* TAG codons are TAGA, including 11/21 of those highly represented in the mass-spec results (Supplementary Table 1). Therefore, it is not surprising we did not see any increase in specificity for a target incorporation site in whole-cell lysate (Supplementary Figure 4D). There are other potential avenues to prevent proteomic incorporation, which include upregulating RF1 to increase termination rates or limited recoding of the genes which we found to incorporate nsAAs at a high rate.

**Cellular Uptake of nsAAs**. As we were conducting nsAA incorporation experiments in *B. subtilis*, we discovered that a variety of

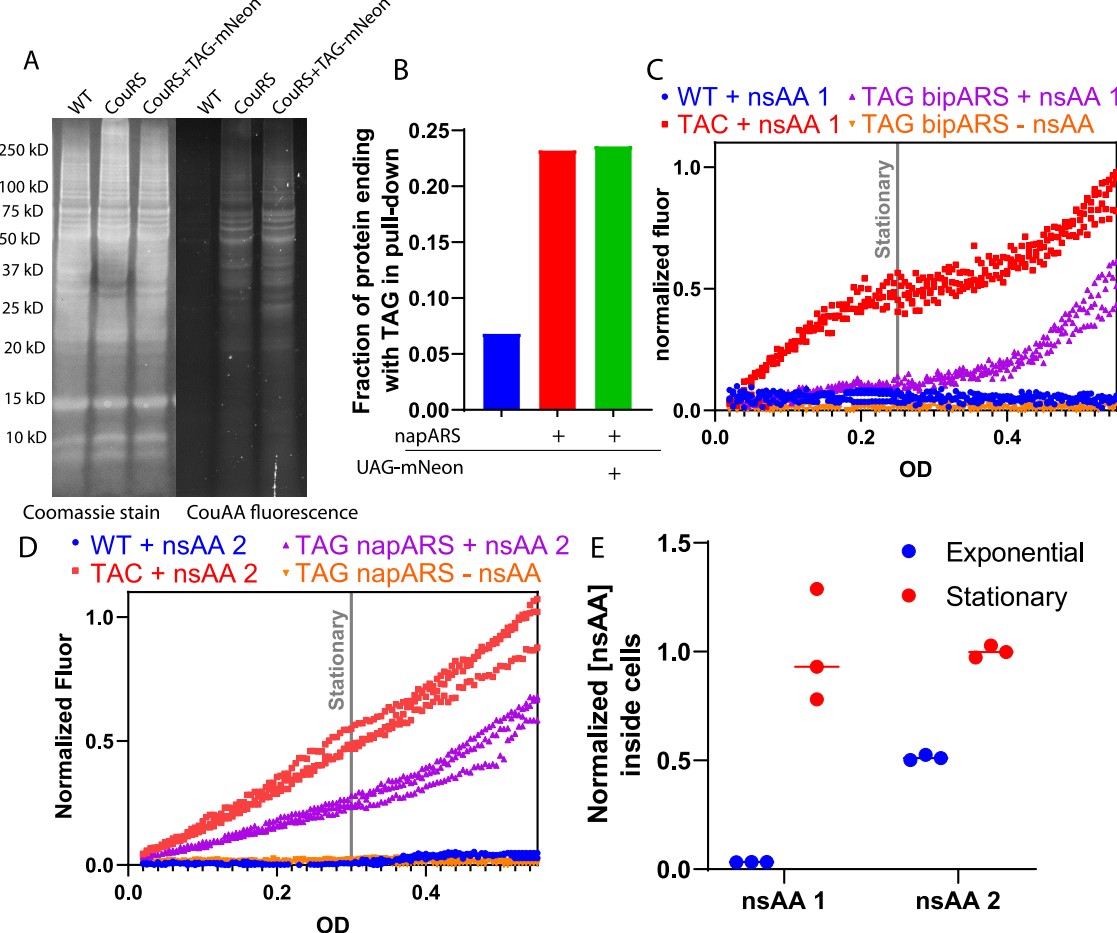

**Fig. 2 Proteomic incorporation and cellular uptake of nsAAs in _B. subtilis_. A** Whole-cell lysate of cells with and without synthetase & cognate tRNA and reporter grown with nsAA **3** run on SDS-PAGE gels. Imaged with Coomassie whole protein stain (left) and fluorescence of nsAA **3** (right) **B**. The fraction of protein sequences whose genes end in TAG from an enrichment for nsAA incorporation. The azide-containing **2** was incorporated by napARS & cognate tRNA, and a click-pulldown performed, after which peptides were detected by mass spectrometry. **C,D** Timecourse of mNeongreen fluorescence vs. OD normalized to maximal TAC-mNeongreen signal of **C 1** and **D 2** incorporation in the presence and absence of corresponding synthetases. 3 biological replicates are shown. Grey vertical lines indicate approximate start of stationary phase. (See Supplementary Fig 5 for corresponding growth curves). **E** Relative concentration of nsAAs inside cells grown to indicated growth phase before being washed and lysed, then internal concentration of nsAAs **1** & **2** measured by LCMS. Values are relative, normalized to OD and to concentration in cells during stationary phase. 3 biological replicates are shown, with a horizontal bar at the median.

factors seemed to influence nsAA transport that may not play a role in _E. coli_, including cell state, media richness, and the specific nsAA used. These findings constrain what nsAAs can be used to study certain phenomena. In rich media, nsAA incorporation is significantly delayed for both **1** and **2** (Supplementary Fig. 5 G–H, 6 G–H). Furthermore, in standard S750 minimal media containing glucose and glutamate, **1** incorporation was delayed until the onset of stationary phase (Fig. 2C & Supplementary Fig. 5A, 6A). For the smaller, more hydrophilic nsAA **1**, equal incorporation was observed in exponential, and stationary phases (Fig. 2D & Supplementary Fig. 5B, 6B). We also demonstrated that **2** could be incorporated during sporulation (Supplementary Fig. 8). nsAA **4**, another bulky hydrophobic amino acid, also showed limited incorporation in the exponential phase, but the pyrrolysine analog **5** was able to incorporate in the exponential phase (Supplementary Fig. 5 C–D, 6 C–D). We hypothesized that nsAA uptake into the cell was limited and inhibited by high concentrations of standard amino acids. Pluronic F-68, a surfactant shown to be non-toxic and to help bulky molecules cross the cell membrane[43] increased **1** incorporation, as did removing all amino acids from the growth media (Supplementary Fig. 5 E–F, 6 E–F).

To verify that import into the cell was limiting, we performed LCMS experiments measuring internal nsAA concentrations under different conditions. We noted that **1** was only present in very low relative concentrations in cells during the exponential phase while **2** was present at higher concentrations inside cells during the exponential phase (Fig. 2E). These findings confirm that the available pool of nsAAs inside the cell is limited for incorporation of **1** during the exponential phase, suggesting potential problems for use of other large, hydrophobic nsAAs in _B. subtilis_. Therefore, future work using nsAAs in _B. subtilis_ may benefit from using smaller and more polar nsAAs if larger and bulkier nsAAs fail to be imported.

**Photocrosslinking.** A primary application of nsAAs is the incorporation of UV-photo-crosslinkers to probe protein structure and assembly in vivo. Previous work has shown that the YukE protein, a homolog of the mycobacterial virulence factor EsxA, requires homodimerization for efficient translocation by the Early secretory antigen (Esx) pathway[44]. We used the photocrosslinking capabilities of **2** to demonstrate short-range specific crosslinking between YukE monomers in _B. subtilis_ cells.

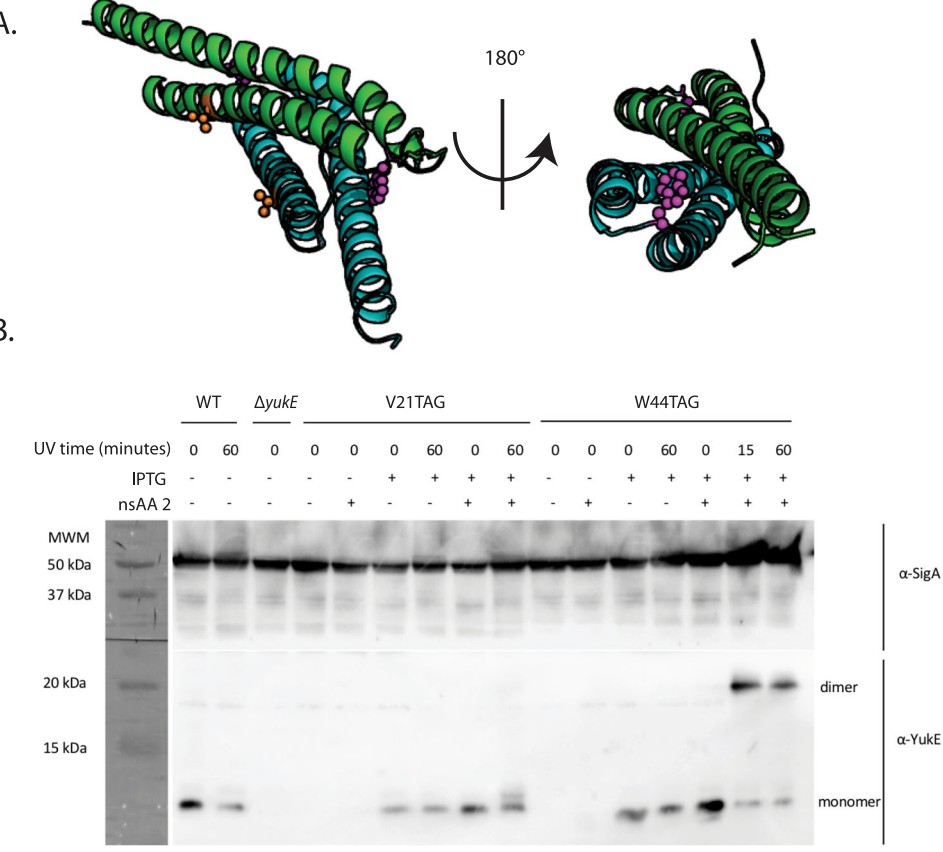

**Fig. 3 In vivo photocrosslinking. A** Positions of V21 (orange space fill) and W44 (purple space fill) indicated. **B** Immunoblotting results from in vivo crosslinking experiment. Cells producing YukE with **2** incorporated at either position 21 or 44 were treated with UV for indicated times. Top panel anti-SigA loading control immunoblot. Bottom panel anti-YukE immunoblot with monomeric and dimeric YukE positions noted.

Consistent with its use in *E. coli*, **2**-based photocrosslinking is capable of distinguishing short-range interactions, as placing the photocrosslinking nsAA on the interface of the homodimer (W44TAG) resulted in high-efficiency crosslinking, while placing it on the external face (V21TAG) did not yield any detectable crosslinking (Fig. 3). The crosslinking efficiency was noted to be extremely high, with nearly 100% crosslinking achieved with only 15 min of UV exposure. The remaining monomer may have resulted from translational readthrough, judging from the control samples lacking nsAA **2**. The exceptional crosslinking efficiency suggests a very tight interaction between the YukE monomers inside cells and supports previous work that suggests in vivo dimerization is required for the export of the YukE dimer[44].

**Translational titration with nsAAs.** Based on our work with **2** for studying cellular uptake and photo-crosslinking, we recognized that this nsAA may allow fine-tuned control of protein expression at the translational level by requiring incorporation for full-length protein formation. Titration of **2** across two orders of magnitude controlled the translation of mNeongreen reporter over a wide linear range and a dynamic range of approximately 50-fold (Fig. 4A and Supplementary Fig. 7A), roughly twice that previously observed for nsAAs in *B. subtilis*[22]. Notably, while most transcriptionally inducible promoters show cooperative-like behavior and have steep induction curves with high Hill coefficients, nsAA-based translational control shows little to no cooperativity, with a Hill coefficient of a similar value to 1 (Table 1).

In *E. coli*, '2-dimensional' regulation of transcription and translation via a combination of inducible promoters and site-specific

nsAA incorporation respectively has been well-demonstrated[45]. *B. subtilis* would particularly benefit from this kind of regulation to achieve highly titratable and zero-leakage expression systems, (Fig. 4B and Supplementary Fig. 7B), which allow 570-fold induction when using **2** and 1300-fold induction with a lower maximum expression when using **5** (Supplementary Fig. 1G). Thus, these strategies are well-suited for experiments where leaky expression or limited dynamic range are barriers to obtaining insights.

One important phenomenon where improved ability to titrate protein concentration would facilitate biological insight is bacterial cell division. This process is orchestrated by the tubulin homolog FtsZ, which forms filaments that treadmill around the cell and assemble into the cytokinetic "Z-ring"[26]. Recent work has demonstrated the importance of FtsZ dynamics in bacterial cell division[12,46]; modulating these dynamics is thus an attractive experimental target. A good candidate for this is the expression of MciZ, a small protein that inhibits FtsZ polymerization, increases its treadmilling dynamics[27], and decreases its filament length[12]. Although the effects of varying levels of MciZ on FtsZ have been characterized in vitro and simulated in silico[47], in vivo experiments have been complicated by FtsZ's high sensitivity to low levels of MciZ[27], requiring a finely controlled expression system. Experiments with standard promoters have been unable to examine this low-expression regime, as even low-level induction prevents Z-ring assembly[27].

Using a pHyperspank-TAG-MciZ construct, we were able to finely control MciZ expression to modulate cell division in live *B. subtilis* cells. We observed that FtsZ filament dynamics & Z-ring assembly were altered with increasing MciZ expression (Fig. 4C, Supplementary Video 1). To quantitatively characterize the

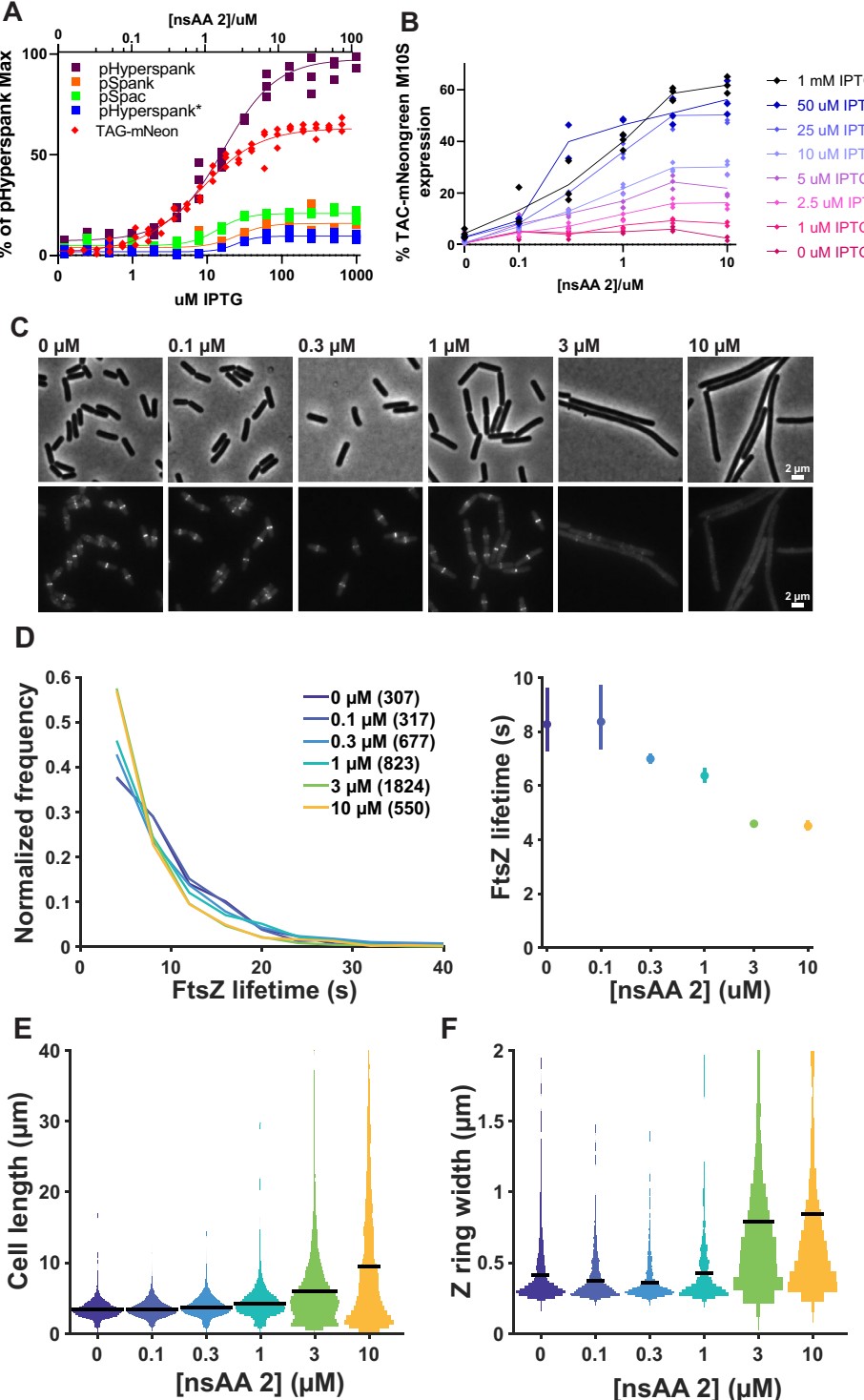

response of FtsZ filaments to MciZ titration, we assayed the single-molecule lifetimes of individual FtsZ filament subunits. The subunit lifetime reflects both the filament length and the rate of filament treadmilling[48]. Based on previous in vitro studies[27], we expected MciZ expression to increase FtsZ dynamics and decrease filament length, both of which should lead to a measurable decrease in FtsZ lifetime. Indeed, increasing MciZ expression caused FtsZ lifetimes to decrease in a dose-dependent manner. This demonstrated that we can precisely modulate FtsZ lifetime by controlling MciZ expression with nsAA translational titration (Fig. 4D).

Next, we investigated the effects of different MciZ levels on the process of cell division. As expected[27], high levels of MciZ expression inhibited Z-ring formation and cell division, as evidenced by increasing cell length (Fig. 4C, D). Notably, we were also able to find low-level induction conditions in which cells were still able to divide, conditions not previously accessible in vivo due to the leakiness and high cooperativity of standard inducible promoters[27]. We found that, although FtsZ subunit lifetimes are reduced even at low levels of MciZ induction (Fig. 4D), Z-ring formation is only disrupted at higher MciZ inductions (Fig. 4E, F). The sharp switch in Z-ring assembly seen

**Fig. 4 Titration of protein concentration using nsAAs. A** Comparison between titration curves of standard IPTG promoters and nsAA titration with nsAA 2. All signals from mNeongreenM10S constructs. The TAG-mNeongreenM10S construct is expressed by a pHyperspank promoter with 1 mM IPTG under all nsAA concentrations. Hill coefficients and dynamic ranges of sigmoidal curves shown can be found in Table 1. All values show 3 replicates with a line representing a sigmoidal fit. **B** 2-dimensional titration of TAG-mNeongreen with both IPTG and nsAA 2. All values are average of 3 replicates, errors are standard deviation. **C** Titration of MciZ expression and its effects on cell division in vivo. Cell morphology (phase contrast, top) and Z-ring morphology (epifluorescence images of mNeonGreen-FtsZ, bottom) in cells expressing MciZ under nsAA 2-inducible control. The concentration of **2** added is indicated above each pair of panels. In each case, the TAG-MciZ construct was induced with 100 μM IPTG. **D** Titration of FtsZ's filament dynamics by MciZ expression. Left: FtsZ single-molecule lifetime distributions at various MciZ expression levels. FtsZ's lifetime reflects its dynamics, with a shorter lifetime corresponding to faster dynamics. Legend: nsAA 2 concentration (number of particles analyzed) Right: MciZ expression level vs mean FtsZ lifetime, computed from single exponential fits to the data at left. Error bars: 95% confidence interval. **E** Increasing MciZ expression impairs Z-ring formation. The frequency of Z-rings along the length of cell was measured in each condition. A control strain lacking the inducible MciZ system is included for comparison. Sample size was $N$ = 7568, 8478, 9886, 11364, 7852, 1605 cells for 0, 0.1, 0.3, 1, 3 and 10 μM of nsAA 2, respectively. **F** Increasing MciZ expression causes aberrant Z-ring assembly. Z-ring width along the length of the cell was measured in each condition as a proxy for successful Z-ring condensation. Z-rings in 3 μM & 10 μM nsAA 2 conditions appear faint and decondensed, but computational analysis of many cells was able to detect Z-rings for both conditions. Sample size was $N$ = 2533, 1338, 737, 533, 447, 485 Z-rings for 0, 0.1, 0.3, 1, 3 and 10 μM of nsAA 2, respectively.

**Table 1 Quantitative characteristics of different promoters in _B. subtilis_.**

| Promoter | Inducer | Dynamic Range | Leakiness (absolute) | Hill coefficent | 95% CI on Hill coefficent |
|---|---|---|---|---|---|
| pHyperspank | IPTG | 13.7 | 7.1 | 1.22 | 1.01 to 1.48 |
| pSpank | IPTG | 4.1 | 3.6 | 2.28 | 1.67 to 3.30 |
| pSpac | IPTG | 4.3 | 4.5 | 2.33 | 1.15 to 6.45 |
| pHyperspank* | IPTG | 5.4 | 1.7 | 3.27 | 2.09 to 7.12 |
| pXyl | Xylose | −10 | 11 | Could not saturate | Could not saturate |
| TAG | 2 | 49.9 | 1.3 | 1.08 | 0.97 to 1.2 |

Characteristics were plotted by fitting sigmoidal curves to a titration, shown in Fig. 4A. Leakiness is measured as % of pHyperspank max expression. pXylose fit shown in Supplementary Figure 7A, and could not be saturated, even with 3 M xylose.

at 3 μM **2** indicates that there is a critical point beyond which FtsZ filaments become too short and/or too dynamic to properly assemble a Z-ring, corresponding to a mean FtsZ lifetime between 4 and 6 s. These observations are consistent with a previous proposal that FtsZ filaments might need to be a minimal length to form into a Z-ring[27]. These experiments use nsAA titration of MciZ expression to precisely modulate FtsZ filament dynamics in vivo, thereby interrogating properties underlying bacterial cytokinesis.

## Discussion

Here we demonstrate general nsAA incorporation tools in the gram-positive model organism _B. subtilis_ and apply them to perform photocrosslinking and tight translational titration. We developed a general and extensible framework for tyrosine, pyrrolysine, and tryptophan-derived nsAAs in this gram-positive organism. These three families constitute 85% of demonstrated nsAAs incorporated in any organism[18], and our characterization of the limitations of nsAAs in this organism demonstrated extensive untargeted incorporation into native _B. subtilis_ proteins, a phenomenon not previously observed in _E. coli_[23–25] or mammalian cells[49]. Additionally, we found that nsAA import can be limiting in rich media or with large, hydrophobic nsAAs, a phenomenon not previously observed in _B. subtilis_ genetic code expansion.

However, despite the proteome-wide incorporation that occurs in _B. subtilis_, we demonstrate both in vivo photocrosslinking and translational titration of protein expression with favorable induction characteristics. Photocrosslinking using nsAAs was extremely high-efficiency, nearing 100% crosslinking using the short-ranged crosslinker **2**. We used this technique to confirm a dimerization interface of the secreted virulence factor YukE, supporting previous studies that suggest dimerization happens inside of cells and is required for export and functionality[44]. Our use of translational titration adds nsAA-dependent protein expression to the toolkit of expression control systems available in _B. subtilis_. The nsAA-driven titration has very low cooperativity and leakiness, and the overall gain can be varied by titrating the associated rate of transcription and translation through inducible promoters or varying the ribosomal binding site.

The use of translational titration allowed us to modulate FtsZ's dynamics and therefore _B. subtilis_ cell division. Specifically, we were able to access a previously unseen regime of FtsZ dynamics in vivo, and identify a critical transition in FtsZ function as we perturb these dynamics. Our data suggest that there is a minimum lifetime of FtsZ in the filament of 5 ± 1 s, below which cell division is impossible because FtsZ filaments are unable to form a functional Z-ring. These results demonstrate that enabling genetic code expansion tools in _B. subtilis_ will not only allow the use of nsAA-based techniques developed in _E. coli_ but also serve as the foundation for developing technologies taking advantage of _B. subtilis's_ status as a gram-positive model, industrial, and Generally Regarded As Safe (GRAS) organism.

We discovered that AARSs that work in _E. coli_ seem to usually function in _B. subtilis_ with similar activity and range of substrates. Because of this portability of _E. coli_ genetic code expansion techniques to _B. subtilis_, future developments to _E. coli_ nsAA tools will likely directly transfer to _B. subtilis_, enabling applications and uses. However, future work in _B. subtilis_ must be cautious of the innate differences between the two systems. First, there appear to be innate differences in the translational termination machinery that allows efficient nsAA incorporation via amber suppression with minimal engineering of the translational machinery but also allows incorporation of the nsAA throughout the proteome. It is possible this is due to overexpression of amber-suppressing tRNA but we think this possibility is unlikely as our approach used a single-copy genomic tRNA, compared to the high-copy number plasmid tRNAs that are driven by strong promoters as reported in many _E. coli_ papers. Though _B. subtilis_ is known to decouple transcription and translation[41] and relax

translational termination during stress[40], further investigation of this phenomenon may uncover deeper differences between the gram-positive and gram-negative protein biology. Second, the amino acid uptake machinery is quite different between these two organisms and will be subject to different limitations. Encouraging uptake of large, hydrophobic nsAAs could be accomplished by heterologous expression of a panel of appropriate membrane transporters[50], and minimal protein engineering may be necessary to maximize uptake[51]. Third, B. subtilis genomic insertion and modification is much simpler than in many bacterial species and will lend itself well to interrogating native genes with nsAAs. These findings will aid the expansion of the genetic code in organisms aside from B. subtilis.

Genetic code expansion in B. subtilis provides multiple potential applications in both the experimental and applied spaces. Highly efficient and controlled nsAA incorporation will enable research in B. subtilis allowing fluorescence, titration, crosslinking, and photocontrol[52] of different proteins. Furthermore, genetic code expansion could allow for biocontainment via synthetic auxotrophy[22,53], which could enhance the use of B. subtilis as an engineered probiotic or therapeutic in humans, livestock, and plants[8,54]. Advances in metabolic engineering that enable in vivo synthesis of nsAAs[55] could improve biocatalytic processes[56], enhancing the already capable B. subtilis chassis for further metabolic engineering[57] and protein production[11].

## Methods

**Reagents**. Antibiotics and nsAAs were purchased from Sigma. nsAA stock solutions were prepared in water with minimal base, e.g., 0.3 M KOH to prepare 0.2 M nsAA 4 stock solution, except for Cou stock, which was prepared in 50% DMSO with minimal base. All stock concentrations of nsAA were between 100 and 200 mM.

**B. subtilis media**. $S7_{50}$ media was made in 500 mL, by mixing 50 mL 10X $S7_{50}$ salts (recipe below), 5 mL 100× $S7_{50}$ metals (recipe below), 5 mL 1 Molar glutamate and 10 mL 50% (w/w) glucose together. Made up to 500 mL with $ddH_2O$. Filter sterilized (not autoclaved) and distributed into 50 mL aliquots. Glycerol, sorbitol, or fumarate was substituted 1:1 for glucose for different carbon sources. Other amino acids can be substituted 1:1 for glutamate or use 20% (w/w) ammonium sulfate (final concentration 0.2% + additional 0.1% from 10× salts) for media lacking amino acids.

$10\times$ $S7_{50}$ salts were made in 1-liter aliquots. 104.7 g MOPS (free acid), 13.2 g ammonium sulfate $(NH_4)_2SO_4$, 6.8 grams potassium phosphate monobasic $KH_2PO_4$ were added and buffered to pH 7 with 50% KOH, then made up to 1 L with $ddH_2O$. Media was filter-sterilized, covered with foil, and store at 4 °C. If yellowed, the solution is no longer usable.

$100\times$ $S7_{50}$ metals have final concentrations of: 0.2 M $MgCl_2$, 70 mM $CaCl_2$, 5 mM $MnCl_2$, 0.1 mM $ZnCl_2$, 100 microgram/mL thiamine-HCl, 2 mM HCL, 0.5 mM $FeCl_3$. Added iron last to prevent precipitation. Media was sterilized and store foil-wrapped at 4 °C.

MC media for transformations was made at a 10× stock. The 10× stock has final concentrations of 1 M potassium phosphate pH 7, 30 mM sodium citrate, 20% (w/w) glucose, 220 mg/mL Ferric ammonium citrate, 1% casein hydrolysate, 2% potassium glutamate. Store in aliquots at −20 degrees, and make 1× media during the competence protocol, supplementing the 1× MC with 3–30 mM $MgSO_4$.

**Strain construction**. All B. subtilis strains were derived from the prototrophic strain PY79[58]. Strains for each figure, strain crosses, strain construction, and primers can be found in a Supplemental excel file. Primers used can also be found in supplemental Table 2. All cloning was done via a combination of Gibson isothermal assembly and overlap-extension PCR. Primers were ordered from IDT with Q5 Tms (calculated via the NEB Tm calculator) of 70–72 degrees. Overlaps between parts were 20–35 bases, attempting to maintain overlap Tm of 55–65 degrees as calculated by the Geneious Tm calculator. PCRs were carried out in 25 μL, using 2× Q5 master mix with the maximum allowed temperature at 25 cycles and with 1 min/kB extension time. PCR fragments were gel-verified and either gel-extracted or PCR-purified using appropriate Qiagen kits. Isothermal reactions were done in 20 μL final volume homemade Gibson mix made according to the original 2011 Gibson paper[59], at 50 °C for 30–45 min. The entire volume of the Gibson was transformed directly into B. subtilis by the B. subtilis transformation protocol described below. In the case of transformation failure, overlap assembly PCR was carried out in a pairwise manner to assemble difficult-to-Gibson regions of DNA. 5–10 μL of each purified PCR product was added to overlap-extension PCRs, which were run for five cycles, followed by the addition of primers

to amplify the full-length piece and 25 more cycles. All constructs were confirmed by sequencing.

**B. subtilis transformation protocol**. For transformation of either genomic or linearized DNA, LB plates were streaked with parent strains the day before. Freshly grown (not more than a day old) colonies were inoculated into 1 mL MC in a large glass test tube, with 3–30 μL of 1 M $MgSO_4$ supplemented. Strains were grown in MC at 37 °C in a roller drum for 4 h, at which point culture should be visibly turbid. Then, 200 μL of culture was added to a standard 13 mL culture tube with transformation DNA. An entire 20 μL Gibson reaction or 2 μL of B. subtilis genomic DNA was added to each tube. Cultures were returned to the 37-degree roller for two more hours, and then the entire volume was plated on selective LB plates. Single colonies were picked and verified by sequencing of unpurified colony PCR. To do colony PCR with B. subtilis colonies, a colony was suspended in 50 μL TE + 10% CHELEX beads, then vortexed for 10 min, boiled for 30 min, vortexed for 10 min, and spun down. 1 μL was used as a PCR template without getting any CHELEX beads into the PCR reaction.

**B. subtilis genomic DNA preparation**. For Bacillus genomic DNA prep, used as a PCR template or to transform into other strains, 1–2 mL LB was inoculated from a fresh colony plated the night before. Cells were grown until dense, preferably not overgrown. Cells were pelleted at max speed and the supernatant aspirated. Cells were resuspended in 500 μL lysis buffer, final concentration 20 mM Tris-HCl pH 7.5, 50 mM EDTA, 10 mM NaCl, and 50 μL of freshly made 20 mg/mL egg white lysozyme in lysis buffer was added. The mixture was incubated at 37 degrees for 15–45 minutes, using the longer time if cells were overgrown. Then 60 μL 10% (w/w) sarkosyl (N-lauroylsarkosine) in $ddH_2O$ was added and the mixture vortexed. The entire solution was transferred to a phase lock tube, and 600 μL phenol-chloroform was added. The tube was vortexed vigorously for 15–20 s until frothy, then spun at max speed for 5 min. The aqueous phase was removed to a fresh 1.5 mL tube and 1/10 volume (60 μL) 3 M sodium acetate added, then vortexed. Two volumes (or top of tube) of 100% Ethanol was added and the tube inverted until DNA had visibly precipitated. Then the tube was spun at max speed for 1 min. The supernatant was aspirated, 150 μL 70% ethanol 30% $ddH_2O$ was added, and the tube quickly vortexed. The tube was then spun for 1 min at max speed. The supernatant was removed, and tubes were left open to dry on a bench for 5–15 minutes. DNA was resuspended in 350 μL $ddH_2O$ and stored at −20 °C.

**nsAA incorporation**. For nsAA incorporation, fresh colonies were picked from LB plates to make a starter in the same media as the experimental media, most often S750 minimal media. These cultures were grown to exponential phase, preferably between OD 0.1 and 0.4, though ODs as high as 0.7 were used without issue. Experimental cultures already containing IPTG and nsAA were seeded from starter cultures at an OD of 0.002. Experimental cultures were grown overnight either in a plate reader or culture tubes at 37 degrees, either in a shaker at 250 rpm or roller drum. For endpoint experiments, cultures were diluted 1:1 with PBS and then read in a Biotek spectrophotometer plate reader, unless the fluorescent nsAA three was being incorporated, in which case cells were pelleted at 5 K rcf and washed 3× with PBS before being read in the plate reader. Fluorescence background signals were corrected for autofluorescence by subtracting the average fluorescence of PY79 cells grown in parallel in media with no additives.

**Click-enrichment of proteins containing nsAAs**. For protein expression of nsAA-containing proteins, 10 mL of S750 culture containing 1 mM IPTG & 100 μM nsAA 2 was seeded at OD 0.002 with appropriate strain from an exponential-phase s750 culture (OD 0.1–0.5). Cultures were grown in a shaking incubator overnight at 37 °C. Cells were pelleted at 5 K RCF for 30 min and frozen at −80. Pellets were lysed with 40×3-second sonication pulses in ice using a QSonica Q125 sonicator using the urea lysis buffer provided by the Thermo Fisher Click-iT™ Protein Enrichment Kit (Cat # C10416). Follow-up purification was performed according to manufacturer's instructions. The click-enriched fraction was trypsin digested off of the beads with 2 μg/mL trypsin in 100 mM Tris, 2 mM CaCl2 buffer & 10% acetonitrile. Dialysis buffer exchange was used to convert the buffer to HPLC solvent A, after which the mass-spec protocol detailed below was followed.

**Protein purification and mass spec**. For protein expression of nsAA-containing proteins, 25 mL of S750 culture containing 1 mM IPTG & nsAA (1 mM for nsAA 3, 5, & 6, 100 μM for nsAAs 1, 2 & 4) was seeded at OD 0.002 with appropriate strain from an exponential-phase s750 culture (OD 0.1–0.5). Cultures were grown in a shaking incubator overnight at 37 °C. Cells were pelleted at 5 K RCF for 30 min and frozen at −80. Cells were lysed using the EMD Millipore Lysonase BugBuster kit (Cat. # 71370-3), following the manufacturer's instructions for gram-positive bacteria. For His-tag purification, Thermo Scientific HisPur cobalt resin (Cat. # 89964) was used. After washing with equilibration buffer, the lysate was bound to resin with 45-minute binding at room temperature followed by one wash step and 3 elution steps, using the following buffers: Equilibration buffer: 20 mM Tris-HCl, Ph 8.3, 0.5 M NaCl, 5 mM imidazole. Wash buffer: 20 mM Tris-HCl, Ph 8.3, 0.5 M

NaCl, 20 mM imidazole. Elution buffer: 20 mM Tris-HCl, Ph 8.3, 0.5 M NaCl, 200 mM imidazole.

Elutions were combined, and 18 µL was run out on an Invitrogen 4–12% Bis-Tris NuPAGE gel (cat # NP0322PK2) following manufacturer's instructions. Purified tagged mNeongreen was seen at the correct weight and excised in approximately 5 × 20 mm pieces.

Excised gel bands were cut into approximately 1 mm³ pieces. Gel pieces were then subjected to a modified in-gel trypsin digestion procedure. Gel pieces were washed and dehydrated with acetonitrile for 10 min, followed by removal of acetonitrile. Pieces were then completely dried in a speed-vac. Rehydration of the gel pieces was with 50 mM ammonium bicarbonate solution containing 12.5 ng/µl modified sequencing-grade trypsin (Promega, 182 Madison, WI) at 4 °C. After 45 min, the excess trypsin solution was removed and replaced with 50 mM ammonium bicarbonate solution to just cover the gel pieces. Samples were then placed in a 37 °C room overnight. Peptides were later extracted by removing the ammonium bicarbonate solution, followed by one wash with a solution containing 50% acetonitrile and 1% formic acid. The extracts were then dried in a speed-vac (~1 h). The samples were then stored at 4 °C until analysis. On the day of analysis, the samples were reconstituted in 5–10 µl of HPLC solvent A (2.5% acetonitrile, 0.1% formic acid). A nano-scale reverse-phase HPLC capillary column was created by packing 2.6 µm C18 spherical silica beads into a fused silica capillary (100 µm inner 191 diameter x ~30 cm length) with a flame-drawn tip. After equilibrating the column, each sample was loaded via a Famos auto sampler (LC Packings, San Francisco CA) onto the column. A gradient was formed, and peptides were eluted with increasing concentrations of solvent B (97.5% acetonitrile, 0.1% formic acid). As peptides eluted, they were subjected to electrospray

One hundred and ninety-five ionization and then entered into an LTQ Orbitrap Velos Pro ion-trap mass spectrometer (Thermo Fisher Scientific, Waltham, MA) controlled by XCalibur 4.3 (Thermo Fisher Scientific, Waltham, MA) software. Peptides were detected, isolated, and fragmented to produce a tandem mass spectrum of specific fragment ions for each peptide. Peptide sequences (and hence protein identity) were determined by matching protein databases with the acquired fragmentation pattern by the software program, Sequest (Thermo Fisher Scientific, Waltham, MA). All databases include a reversed version of all the sequences, and the data were filtered to between a one and two percent peptide false discovery rate

**LCMS nsAA import quantification.** Protocol adapted from Reference 60[60]. To assay import of nsAAs into cells, serial dilutions of WT Py79 *B. subtilis* were seeded in 1 mL S750 media tubes and incubated at 37 °C in a tube roller overnight to obtain turbid but not stationary cultures, OD 0.4–0.8. In the morning, 200 µL of this culture was added to 60 mL experimental cultures containing nsAA. After growth in a 37° shaker until OD ~0.2, exact OD's were recorded and 50 mL of culture taken, the rest left in the 37 ° shaker overnight. 50 mL of culture was pelleted at 4 °C for 10 min at 5.25 K RCF, and the supernatant was discarded. The pellet was washed 4 times with 1 mL ice-cold s750, with rapid but thorough resuspension and 2.5-min spins at 14 k RCF. Cell pellets were then frozen. After the overnight, the saturated culture was diluted 10:1 for OD measurement, and then 5 mLs of saturated culture was pelleted at 4 °C for 10 min at 5.25 K, followed by 4 washes with 1 mL ice-cold s750, with rapid but thorough resuspension and 2.5-minute spins at 14 k RCF and brief freezing. To thoroughly lyse cells, pellets were resuspended in 400 µL 40:60 sterile methanol:water and transferred to screw-top tubes. 300 mg of 200 µm acid-washed glass beads were added to each tube, and the tubes were beaten in a bead beater for 10 min at 4 °C, in 1-min increments with a 5-min gap between each bead beating step. The tubes were inverted and a small hole poked in the bottom with a syringe. Then the screw-top tubes were placed into 7 mL tubes, and the lysate was collected by a 5-min spin at 4000 rcf. An additional 400 µL 40:60 methanol was added to the tubes, and the spin was repeated to ensure all lysate was collected. The collected lysate and wash were spun at 16 k rcf at 4 °C for 30 h, and 750 µL was transferred to fresh tubes, which were then centrifuged at 4 °C for 2 h, and 600 µL supernatant extracted to be used in LCMS experiments.

**nsAA titration microscopy.** Two days prior to imaging, cells were struck from a glycerol stock at −80 °C onto an LB agar plate, and grown overnight at 37 °C. The following day, single colonies were inoculated from the LB plate into 1 mL S750 media containing 100 µM IPTG and the appropriate concentration of nsAA 2. A 1:10 dilution series was prepared out to 1:1,000, and all of these cultures were grown overnight at room temperature on a rolling drum. The next day, the culture in the dilution series that was nearest to mid-exponential phase growth was diluted 1:10 and grown in 1 mL S750 + IPTG + nsAA 2 at 37 °C on a rolling drum. For single-molecule lifetime imaging, 20 pM JF549-HaloTag Ligand was additionally added to the media. Once the density of this culture reached mid-exponential phase, cells were ready to image. Cultures were pelleted by centrifugation for 2 min at 7000 rcf. Approximately 900 µL of the supernatant media was removed; this volume was adjusted depending on the exact OD of cells in the culture. Cells were resuspended in the remaining media.

For imaging, 2 µL of these concentrated cells were pipetted on to a glass coverslip, and immobilized underneath a small pad of S750 agarose. Agarose pads were molded using 1.5 cm×1.5 cm×1 mm plastic frames, which were placed on a pane of glass that had been cleaned with detergent and 70% ethanol. Molten S750 + 2% agarose was poured into the frames, and a second glass plane was placed

on top to complete the mold. Pads solidified at room temperature for at least 15 minutes, and excess agarose was cut away from the outside of the frame prior to use. Microscopy was performed on a Nikon Ti-E microscope equipped with TIRF optics, a perfect focus system (Nikon), and a MLC4008 laser launch (Agilent). The objective used was a Nikon CFI Plan Apochromat DM Lambda 100X oil immersion objective with 1.45 NA and a Ph3 phase-contrast ring. The microscope was enclosed in a chamber heated to 37 °C. Images were collected using NIS-Elements software (Nikon).

Imaging of FtsZ filaments and Z-rings and quantification of Z-ring morphology was performed using strain bGS543, which contains the napARS system under the constitutive pVeg promoter and TAG-MciZ under the IPTG-inducible pHyperspank promoter. To visualize FtsZ, the strain contains a duplicate copy of the FtsAZ operon, including its promoter, in which FtsZ has been tagged with mNeonGreen on its N terminus. A 488 nm laser was used for excitation and a C-NSTORM QUAD filter (Nikon) and a ET525/50 m filter (Chroma) were used for emission. Images were collected on an ORCA-Flash4.0 V2 sCMOS camera (Hamamatsu). TIRF time lapses were taken with 1 s exposures for 4 min total; after each time lapse, an epifluorescence image was taken to visualize Z-ring morphology and a phase-contrast image was taken to visualize cells. To identify Z-rings in the image, cells were segmented using DeepCell to generate binary masks. The pill mesh function in Morphometrics was then used to generate midlines down the long axis of each cell. Using custom MATLAB code available in the code availability section, the fluorescence intensity of FtsZ was averaged along each cell midline by taking the average along each mesh spline, these intensity traces were smoothed, and Z-rings were identified by peak detection. To plot the Z-ring width distributions, the full width at half maximum of each Z-ring peak was calculated. To measure Z-ring width, a 1-µm region around each peak was subselected from each intensity trace, and these traces were averaged to create an average intensity trace. The Z-ring width was measured by calculating the full width at half maximum of the Z-ring peak in these intensity traces[48].

Single-molecule lifetime imaging and cell length measurements were performed using strain bGS545, which contains the napARS system under the constitutive pVeg promoter and TAG-MciZ under the IPTG-inducible pHyperspank promoter. To visualize FtsZ, the strain contains a duplicate copy of the FtsAZ operon, including its promoter, in which FtsZ has been tagged with HaloTag on its N terminus. The HaloTag-FtsZ construct is labeled with JF549-HaloTag Ligand for fluorescence microscopy. TIRF time lapses were taken using 500 ms exposures for 4 min total; after each time lapse, a phase-contrast image was taken to visualize cells. A 561 nm laser was used for excitation and a C-NSTORM QUAD filter (Nikon) and a ET600/50 m filter (Chroma) were used for emission. Images were collected on an iXon Ultra 897 EMCCD camera (Andor). To analyze the data, the phase images of cells were first segmented using DeepCell to generate cell masks. Next, TrackMate was used to identify single particles in the image and preliminarily link them together. Spots were detected with a 1.5-pixel radius and an intensity threshold that was manually selected for each data set. Spots were then linked roughly into tracks, with a three-pixel linking distance and a maximum gap of ten frames; in this way, only localizations with at least one other spot detected nearby were considered for further analysis, which decreased computational load in the next stage. These data were exported into MATLAB for further analysis. The tracklist from TrackMate was then filtered and converted to intensity traces. First, the spot positions in each track were averaged to generate a mean position of each spot. Next, spots that were not inside cells were excluded using the cell masks generated by DeepCell. Spots within three pixels of one another were then combined, and a new average position was calculated, weighted based on the length of each track. Then, for each spot, an intensity trace was generated: intensity was averaged in a 5 × 5-pixel window around the mean spot position, and the local background was averaged in a 2-pixel frame around the window and subtracted to generate a background-subtracted trace. Finally, intensity traces were smoothed; only traces with a maximum background-subtracted intensity above 500 counts were included for further analysis. To measure each single-molecule lifetime, these intensity traces were fit to a hidden Markov model using the MATLAB package vbFRET. To rule out spots that contained no single-molecule fluorescence events, and to exclude cases in which multiple single molecules overlap, models were fit with one, two, three, and four states. Bayesian model selection was used to select the best-fitting model, and only traces in which the two-state model fit best were included. Traces for which the difference between state 1 (no fluorescence) and state 2 (single-molecule fluorescence) was less than 60 counts were also discarded. The duration of each state-2 event was measured; dwell times less than 2 s (four frames) were discarded, as were events that overlapped with the start or end of the trace since they cannot be measured accurately. Traces containing more than two events were also excluded. The resulting single-molecule lifetimes were fit to a single exponential distribution and the mean lifetime was computed. We measured the contribution of photobleaching to our lifetimes by repeating the experiment at 1-s imaging intervals rather than 500-ms intervals without changing the exposure time; the measured lifetime did not change, indicating that the photobleaching contribution was negligible. To measure cell lengths, cell images were segmented using DeepCell and their dimensions were measured using Morphometrics Further analysis was performed with custom code available in the code availability section. Cells that extended out of the field of view were excluded from the analysis because their lengths could not be accurately measured; however, at the highest levels of MciZ expression, some cells were too long to be captured in a single field of view.

Thus, these cell length measurements underestimate the true cell length for samples induced with 3 and 10 μM nsAA 2.

**Incorporation of unnatural amino acids during sporulation**. Sporulation was induced by resuspension at 37 °C. OD 1 cultures were centrifuged and resuspended in the same volume of SM media. One liter of SM media is made by adding 46 mg FeCl₂, 4.8 grams MgSO₄, 12.6 mg MnCl₂, 535 mg NH₄Cl, 106 mg Na₂SO₄, 68 mg KH₂PO₄, 96.5 mg NH₄NO₃, 219 mg CaCl₂, 2 g L-glutamic acid, 20 mg L-tryptophan. with the following modifications. Where appropriate, at the time of resuspension, the inducer, isopropyl β- d-1-thiogalactopyranoside (IPTG), and nsAA 2 was added to the test cultures at 1 mM and 100 μM final concentration, respectively.

Images were collected using an Axio M1 Imager (Zeiss). For strains producing GFP, the GFP channel excitation was 30 milliseconds (Ex bandpass 470 ± 40 and Em - 525 ± 50). Images were analyzed and prepared using Zen 2.0 software (Zeiss) and ImageJ.

**Detection of protein–protein interaction by crosslinking in vivo**. Strains were grown in 30 mL LB medium at 37 °C with shaking at 210 rpm to an OD 600 nm of 1.0–1.5. The cultures were split into separate flasks into 7 ml aliquots. As appropriate, cultures were either supplemented with the inducer (IPTG) or with nsAA 2, to final 1 mM and 100 μM concentration, respectively, or both or neither.

Growth at 37 °C with shaking was continued for 13 h. At this time, OD measurement of cultures was taken again (generally around OD600 3.40–4.9), and 1 mL aliquots were pelleted by centrifugation and resuspended in 1 mL of sterile 1x PBS buffer. Cell resuspensions were transferred to a 12-well polycarbonate plate (Corning, catalog # 353043) and exposed to UV light at 365 nm in Spectrolinker XL-1000 UV Crosslinker (Spectronics Corporation). The distance between the bottom of the well with cell resuspension and the UV source was ~4 cm. Cell suspensions were exposed to the UV source for either 15 min or a total of 60 min, which was divided in two 30-min intervals. 12-well plate was on ice, and thermal probe readings confirmed that the cell suspension temperature never exceeded 35.1 °C.

After UV treatments, cell suspensions equivalent to 1.5 OD units were pelleted by centrifugation, Cell pellets were resuspended in 100 μL of Tris-HCl pH 6.8 lysis buffer, containing 10 mM EDTA, 100 μg/mL chicken egg white lysozyme, 10 μg/mL DNaseI, 0.25 mM phenylmethylsulfonyl fluoride (PMSF) and 1 μM protease cocktail inhibitor E-64. Lysis was continued for 20 min at 37 °C, and then 50 μL SDS-PAGE loading buffer (without reductant) was added. The lysed samples were mixed vigorously and then heated for 10 minutes at 65 °C.

For crosslinking detection, 7.5 μL aliquots of all control and test samples were separated by 17% SDS-PAGE. Then proteins were transferred from the gel to polyvinylidene fluoride (PVDF) membrane for 52 min at 15 V in Trans Blot Semi Dry Transfer Cell (BioRad). The membrane was cut horizontally along the 25 kDa marker band to allow for probing with two different antibodies. After blocking, the lower part of the membrane was incubated with custom rabbit polyclonal anti-YukE antibodies used in a dilution of 1:5,000. The upper membrane portion (used as loading control) was probed with anti-SigmaA antibodies diluted 1:10,000[61]. The primary antibodies were tagged successively with goat anti-rabbit antibodies conjugated with horseradish peroxidase (HRP) (Abcam, catalog # ab6721) at a 1:3,000 dilution. The bound antibodies were detected using luminol containing reagent Clarity Western ECL Substrate (BioRad) and visualized using ChemiDoc imager (BioRad).

**Statistics and reproducibility**. The gel shown in Fig. 2a was repeated, with variations, on four occasions. On each occasion, the same general pattern of a lack of fluorescence for WT cells or cells lacking the synthetase was noted, while cells with the synthetase to which the fluorescent nsAA 3 was added demonstrated broad proteomic fluorescence.

The nsAA concentration liquid chromatography experiments in Fig. 2e was repeated three times, in general demonstrating the same broad trend of nsAA 2 importing into the cell during exponential while nsAA 1 does not. Issues with the calibration curves prevented these experiments from being presented alongside these data.

Crosslinking experiments were conducted a total of three times with varied crosslinking conditions, confirming crosslinking with a range of crosslinker concentrations and crosslinking times across the replicates. Two rounds of experiments used the fusion protein YukE(W44TAG)-GFP (strain: bKG028 amyE::Phyperspank -yukE(W44TAG)-GFP (spec) lacA::Pveg-napARS-Pser-tRNA:cat yukE:erm-Pyuk). The third round shown in the figure used YukE(W44TAG), as more directly comparable controls from previously published studies were available for this sample (strain: bKG031 lacA::Pveg-napARS-Pser-tRNA::cat amyE::Phyperspank-yukE(W44TAG) (spec) yukE:erm-Pyuk).

The gel shown in Supplemental Fig. 2a was only run once, as a way to size-separate mNeongreen from other proteins that emerged from his-purification. A similar gel was run one additional time with the UAC-mNeongreen, the lane looked broadly similar to that seen in the bipAS + nsAA 1 lane.

The microscopy images shown in Supplemental Fig. 3c are representative of similar experiments, which were repeated approximately ten times. Most such

experiments were attempting to use nsAA 3 incorporation for subcellular protein localization and failed due to the broad cellular fluorescence seen here.

The gel shown in Supplemental Fig. 4d was only performed once.

The microscopy shown in Supplemental figure 8 was performed once.

**Reporting Summary**. Further information on research design is available in the Nature Research Reporting Summary linked to this article.

## Data availability
All the DNA sequence data used in this manuscript is provided in the Supplementary Information files, and all raw numerical data and gel images obtained from measurements in this study are also provided as Supplementary files associated with this manuscript. Source data are provided with this paper.

## Code availability
Custom code is available at https://bitbucket.org/garnerlab/squyres-2020/src/master/

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

## Acknowledgements

We thank Dr. Valerie Pivorunas for construction of strain bBB954. We thank Michaela Jones, Kamesh Narasimhan, Gabriel Filsinger for comments on this manuscript. We also thank S. Wilson, M. Holmes, W. Mallard, S. Hurlimann, M. Dion, M. Baas-Thomas, G. Filsinger, A. Rudolph, P. Smith, and Drs. G. Chao, A. Debnath, M. Shubert, A. Chatterjee, A. Bisson, A. Florez, J. Marchand, A. Mijalis, K. Narasimhan, A. Nyerges, N. Ostrov, D. Thompson, T. Wannier for helpful discussions and advice. The Taplin Mass Spectrometry Facility and the Analytical Chemistry Core at Harvard Medical School were also essential to this work. This work was supported by US Department of Energy Grant DE-FG02-02ER63445 (to GMC), NSF grant MCB-2027074 (To EG & AMK), NIH R01-GM121865 (to BMB), the Landry Cancer Biology Research Fellowship (to DS) and NSF GRFP DGE1144152 (to GRS).

## Author contributions

D.S., G.R.S., A.K. and A.J. were involved in strain construction. D.S performed non-standard amino acid incorporation and growth experiments with help from E.K., A.K. and A.J. D.S. and E.K. purified proteins and performed biochemical experiments. D.S. and J.R. performed cell import assays. G.R.S. carried out microscopy experiments and analyzed data. K.G. and B.B. performed photocrosslinking assays and analysis. B.B., A.K., E.G. and G.C. supervised the research. D.S., A.K., E.K. and E.G. wrote the manuscript with feedback from all other authors.

## Competing interests

The authors declare the following competing interests: G.M.C. has related financial interests in ReadCoor, EnEvolv, 64-X, and GRO Biosciences. For a complete list of G.M.C.'s financial interests, please visit arep.med.harvard.edu/gmc/tech.html. No other authors declare any conflict of interests.
