## [Peer Review File · Nature Communications]

Reviewers' Comments:

Reviewer #1:

Remarks to the Author:

In this manuscript, Stork et al reported some site-specific non-standard amino acid (nsAA) incorporation systems in the model Gram-positive bacterium, *Bacillus subtilis*.

The key results and my evaluations are as follows:

(1)

The authors demonstrated the incorporation of 20 distinct nsAAs into ribosomally synthesized proteins in *B. subtilis*.

This is the most important result of this paper. Because of high practical utility of *B. subtilis* for both fundamental research and industrial use, this result is useful for wide range of researchers.

On the other hand, the nsAA-specific aaRS-tRNA(CUA) pairs used in this study were originally developed in *E. coli*. It is not surprising that those engineered aaRS-tRNA(CUA) pairs are also functional in *B. subtilis*. Conceptual advance, therefore, is limited in this result.

(2)

The authors found that the nsAAs were incorporated into many proteins across the proteome at UAG sites.

This observation might be an important discovery in this research field. However, the authors need to clarify the following questions and comments described in Major Points to precisely evaluate the significance of this result.

(3)

The authors found that nsAA import can be limiting in rich media or with large, hydrophobic nsAAs.

This information is useful for users. The significance of this result is "moderate".

(4)

The utility of photo-crosslinking and translational titration were experimentally demonstrated. These results are also useful for users. However, these are techniques transferred from *E. coli* to *B. subtilis*. The conceptual advance of these results is limited.

Overall, the manuscript is well written, except for some portions specially mentioned below. Both researchers in closely related fields and general readers will easily understand the paper

<Major points>

(1) page 7-8, "Proteome-wide incorporation of nsAAs at UAG sites": page 18, line 3-8: (related discussion)

The authors emphasize the importance of this finding which may uncover deeper differences between gram-positive and gram-negative protein biology.

However, the analysis and discussion are inadequate and not enough informative to lead readers to believe the authors' expectation. In the present form, thus, I evaluate this finding only a useful tip of experiment although this observation might be an important discovery in this research field. I encourage the authors to describe all possible mechanisms underlying this finding and having a deeper discussion. For example, it might be worthy discussing the following questions.

"The nsAA incorporation at natural UAG stop codons has also been reported in *E. coli* (e.g., Aerni et al, *Nucleic Acids Research* 43(2), e8, 2015). Some strongly expressed genes, which contain a UAG translation termination signal, may exist in *B. subtilis* although UAG is predominantly found in genes expressed at low levels in *E. coli*. Are there really essential difference in the translational termination mechanisms between Gram+ and - bacteria?"

"May over-expression of tRNA(CUA) cause the off-target incorporation?"

Fig 2B suggested that only less than 25% of off-target incorporation was derived from proteins

ending with TAG. On the other hand, the authors described that 23 of the 569 UAG-containing proteins made up 71% of the observed proteomic incorporation events" (page 7, line 16-17, Suppl Table 1) . Are all off-target incorporations explainable by the incorporation at the natural UAG?

(2) page 9-10, "Cellular Uptake of nsAAs"

Page 10, line 17-18; "suggesting problems for other large, hydrophobic nsAAs. As a result, in the near-term small and polar nsAAs can be used for a broader range of studies in *B. subtilis*." The authors tested only 4 nsAAs. This conclusion may be over generalization and should be reconsidered. Otherwise, all nsAAs successfully incorporated in this study should be additionally tested.

(3)page 13, line 4-6 "Because the transcriptionally inducible promoters act by controlling mRNA levels, and the UAG titration controls translation, it is possible to utilize both concurrently for '2-dimensional titration'"

The extremely tight control of gene expression using the transcription-translation dual regulation, in which both inducible promoters and site-specific nsAA incorporation, have already been proposed and experimentally evidenced.

1. Minaba M, Kato Y. (2014) High-yield, zero-leakage expression system with a translation switch using site-specific unnatural amino acid incorporation. *Appl. Environ. Microbiol.* 80(5), 1718-1725. No literature is cited in this part of the paper, which may mislead readers into thinking that the concept which the authors named as "2-dimentional titration" is a completely original idea of the authors. The above papers should be cited and discussed here.

<Minor points>

Page 6, Figure 1C

akbRS should be defined. Is this identical to MaPyIRS?

Page 6, Figure 1C-D

The values of leakage and gain should be indicated to evaluate the utility of nsAA incorporation systems. The background level of leakage in the parent strains, which do not have nsAARS/tRNA(CUA), should also be shown.

page 7, line 8-9 "synthetase expression alone resulted in cellular retention of fluorescent nsAA (Supp. Fig. 4A)."

The cognate tRNA(CUA) may be co-expressed.

Page 7, line 12-13 "off-target nsAA incorporation"

Page 7, line 22; "misincorporation"

These terms should be defined.

Page 9, Fig 2 and others

The identity of nsAA is indicated as their names (e.g., bipA, CouAA and pAzF). The compound number defined in Fig 1B (e.g., nsAA 1) is also used in other figures and text. The authors should use only either of them.

Page 10, Line 2 "2 incorporation was delayed"
nsAA 1 may be correct.

Page 10, line 3 Fig. 2D

Figure 2C may be correct.

Page 10, line 4; Fig. 2E

Fig. 2D may be correct.

Page 10, line 5; "1 could be incorporated during sporulation"

nsAA 2 may be correct.

Page 14, Figure 4A

The promoter driving TAG-mNeon should be indicated.

Page 14, Figure F

Z-ring cannot be seen at 3-10 μ M nsAA 2 in Figure 4C.

Page 17, line 13-14; "Combining transcriptional and translational titration allowed us to modulate FtsZ's dynamics and therefore *B. subtilis* cell division."

The legend of Fig.4 indicated that the concentration of IPTG was constant (100 μ M), suggesting that the authors did not use the combination of transcriptional and translational titration in this experiment.

Supp Fig. 3

The authors should clarify the following points:

(1) Was the tRNA(CUA) co-expressed?

(2) Was CouAA supplied in all experiments?

(3) The upper panel may be CouRS[+tRNA(CUA)] without TAG-mNeon. Otherwise, what is the "background fluorescence" (page 7, line 7)?

Supp Fig 4A

Was the tRNA(CUA) co-expressed?

What is "TAG" (TAG-mNeon)?

Supp Fig 5F

The OD-fluorescence curve for UAC+bipA, in which the normalized fluorescence was saturated at a lower OD, is largely different from others. The absolute value of bipA intake may be not enhanced.

//

Reviewer #2:

Remarks to the Author:

The manuscript by Stork et al. describes a new (expandable) toolbox for the incorporation of non-standard amino acids (nsAA) in *Bacillus subtilis*. *Bacillus subtilis* being a widely-used model species in a broad range of applications (fundamental and applied), this paper will be of interest to a large readership. Moreover, the authors bring attention on intriguing differences between *Bacillus subtilis* and *E. coli* (in which such tools have been extensively characterised), related to nsAA import and incorporation, in addition to their demonstration that the nsAA incorporation systems (nsAA + tRNA / aminoacyl-tRNA synthetase pairs) previously used in *E. coli* also work in *B. subtilis*. Overall this is a carefully conducted study including lots of interesting data for future work. I particularly appreciated that the authors raised awareness on (and extensively characterized) the limitations of their system. These limitations stem from the finding that nsAA incorporate at the proteome level in a non-specific manner at UAG stop codons in *Bacillus*, implying that *Bacillus* must be more tolerant to proteome-wide amber stop codon suppression than *E. coli*. While no mechanism is investigated here, this idea will probably lead to interesting research paths and may lead to the discovery of major discrepancies between these two models regarding translation (termination) control. This non-specific incorporation hinders specific labelling strategies for now. I found their most interesting contribution being the possibility to drastically improve the dynamic range and tight control of protein levels in the cell by combining transcriptional control (via inducible promoters) and translational control (via controlled nsAA incorporation). The authors provide a proof of concept for this by modulating MciZ (an FtsZ inhibiting protein) levels, which has proven difficult so far due to the high sensitivity of FtsZ to even basal level of MciZ. In addition, they also incorporated UV-crosslinkable nsAA and performed in vivo cross-linking with particularly high efficiency, validating previous known interaction data. I only have minor comments.

Fig 1.

- Panel D: was the signal normalised to the TAC-M10S-mNeongreen? If yes, adapt the Y axis label.

Fig 2.

- A-B: it is unclear from the text or legend if the enrichment was performed on cells carrying the synthase alone as in Suppl Fig 4 or the corresponding tRNA too.

Suppl Fig 1.

- Why are different concentration of nsAA used depending on the nsAA nature?

- Panel A, it is unclear if the mNeonGreen construct in panel A is the UAG-mNeon or UAG-M10S-mNeon. Adapt if needed.

- Legend of panel A indicates normalisation to the max fluorescence from the experiment while the Y axis labels shows 'Fraction of UAC-mNeon fluor', similar to Fig 1D ('% of TAC-mNeongreen'). Please clarify what the normalisation is for Fig S1A and homogenize labels/legends across figures when appropriate.

- Panels C-D: is there a reason for using abkRS in E. coli vs MaPyIRS in Bacillus for incorporating nsAA 5 (boc-K)?

Suppl Fig 2.

- Panel A, please indicate the expected size of mNeonGreen on the gel. There is a mistake in the MW marker labels. What is the small (~17 kDa) band found in all lanes except for bocK-containing mNeonGreen?

- B; It is unclear where the elastin-like peptide is inserted in mNeonGreen.

- Mass spectra and fragmentation patterns (C-H) are very small and impossible to read without zooming. Consider moving them in a spreadsheet.

REVIEWER COMMENTS

Reviewer #1 (Remarks to the Author):

In this manuscript, Stork et al reported some site-specific non-standard amino acid (nsAA) incorporation systems in the model Gram-positive bacterium, *Bacillus subtilis*.

The key results and my evaluations are as follows:

(1)

The authors demonstrated the incorporation of 20 distinct nsAAs into ribosomally synthesized proteins in *B. subtilis*.

This is the most important result of this paper. Because of high practical utility of *B. subtilis* for both fundamental research and industrial use, this result is useful for wide range of researchers.

On the other hand, the nsAA-specific aaRS-tRNA(CUA) pairs used in this study were originally developed in *E. coli*. It is not surprising that those engineered aaRS-tRNA(CUA) pairs are also functional in *B. subtilis*. Conceptual advance, therefore, is limited in this result.

(2)

The authors found that the nsAAs were incorporated into many proteins across the proteome at UAG sites.

This observation might be an important discovery in this research field. However, the authors need to clarify the following questions and comments described in Major Points to precisely evaluate the significance of this result.

(3)

The authors found that nsAA import can be limiting in rich media or with large, hydrophobic nsAAs.

This information is useful for users. The significance of this result is “moderate”.

(4)

The utility of photo-crosslinking and translational titration were experimentally demonstrated.

These results are also useful for users. However, these are techniques transferred from *E. coli* to *B. subtilis*. The conceptual advance of these results is limited.

Overall, the manuscript is well written, except for some portions specially mentioned below.

Both researchers in closely related fields and general readers will easily understand the paper

We thank reviewer #1 for their thoughtful and very thorough evaluation of our manuscript.

<Major points>

(1)page 7-8、”Proteome-wide incorporation of nsAAs at UAG sites”: page 18, line 3-8: (related discussion)

The authors emphasize the importance of this finding which may uncover deeper differences between gram-positive and gram-negative protein biology.

However, the analysis and discussion are inadequate and not enough informative to lead readers to believe the authors' expectation. In the present form, thus, I evaluate this finding only a useful tip of experiment although this observation might be an important discovery in this research field. I encourage the authors to describe all possible mechanisms underlying this finding and having a deeper discussion. For example, it might be worthy discussing the following questions.

"The nsAA incorporation at natural UAG stop codons has also been reported in *E. coli* (e.g., Aerni et al, *Nucleic Acids Research* 43(2), e8, 2015). Some strongly expressed genes, which contain a UAG translation termination signal, may exist in *B. subtilis* although UAG is predominantly found in genes expressed at low levels in *E. coli*. Are there really essential difference in the translational termination mechanisms between Gram+ and – bacteria?"

"May over-expression of tRNA(CUA) cause the off-target incorporation?"

We thank reviewer 1 for informing us of existing literature on nsAA incorporation at native sites in *E. coli*. We agree that more discussion of this topic is warranted and have inserted additional comparisons to our findings. As the cited literature found only two instances of nsAA incorporation into native *E. coli* proteins in a RF1+ strain, we believe that there is still a significant difference between translational termination in gram+ & gram- bacteria, and we have cited more literature supporting this claim.

It is possible that the differences are caused by different levels of expression of tRNA(CUA), but we think this is unlikely as our approach used a single-copy genomic tRNA(CUA), compared to high-copy number plasmid tRNA(CUA)s driven by strong promoters in many *E. coli* papers. We have clarified this point in the results & discussion sections.

Fig 2B suggested that only less than 25% of off-target incorporation was derived from proteins ending with TAG. On the other hand, the authors described that 23 of the 569 UAG-containing proteins made up 71% of the observed proteomic incorporation events" (page 7, line 16-17, Suppl Table 1) . Are all off-target incorporations explainable by the incorporation at the natural UAG?

We apologize for the misrepresentation of Figure 2B. We have clarified that that this experiment is an enrichment for proteins containing nsAA, not a purification or determination of exclusively proteins in which nsAA was incorporated. The makeup of the mass spec in which no AARS/tRNA(CUA) were present cursorily resembles whole-cell proteomics of *B. subtilis*. We have corrected figure 2B, the figure caption and portions of the text to more clearly explain that we see significant enrichment for proteins ending with TAG in the presence of the AARS/tRNA(CUA), but we cannot fully describe the distribution of native proteomic incorporation. Our data does not allow us to say if all off target incorporations are explainable by incorporation at native UAG sites.

(2) page 9-10, "Cellular Uptake of nsAAs"

Page 10, line 17-18; "suggesting problems for other large, hydrophobic nsAAs. As a result, in the near-term small and polar nsAAs can be used for a broader range of studies in *B. subtilis*."

The authors tested only 4 nsAAs. This conclusion may be over generalization and should be reconsidered. Otherwise, all nsAAs successfully incorporated in this study should be additionally tested.

We have corrected the language in the paper to avoid the conclusion that import problems into the cell are necessarily due to size and hydrophobicity. However, we believe we should suggest the reasonable possibility that bulkier nsAAs may cause problems for future work to aid readers in designing their own experiments.

(3)page 13, line 4-6 “Because the transcriptionally inducible promoters act by controlling mRNA levels, and the UAG titration controls translation, it is possible to utilize both concurrently for ‘2-dimensional titration”

The extremely tight control of gene expression using the transcription-translation dual regulation, in which both inducible promoters and site-specific nsAA incorporation, have already been proposed and experimentally evidenced.

1. Minaba M, Kato Y. (2014) High-yield, zero-leakage expression system with a translation switch using site-specific unnatural amino acid incorporation. Appl. Environ. Microbiol. 80(5), 1718-1725.

No literature is cited in this part of the paper, which may mislead readers into thinking that the concept which the authors named as “2-dimentional titration” is a completely original idea of the authors. The above papers should be cited and discussed here.

We apologize for our lack of discussion of previous literature on this topic. This has been rectified with a citation of the mentioned literature and a brief highlight of the potential usefulness of such systems to *B. subtilis* research.

<Minor points>

Page 6, Figure 1C

akbRS should be defined. Is this identical to MaPyIRS?

abkRS is a popular *M. barkeri* tRNA synthetase commonly used in *E. coli* to incorporate pyrrolysine nsAAs, especially the photocrosslinker AbK, and while it is homologous to MaPyIRS it is not the same enzyme. We have clarified the caption of figure 1 and the text of the manuscript to make this clear.

Page 6, Figure 1C-D

The values of leakage and gain should be indicated to evaluate the utility of nsAA incorporation systems. The background level of leakage in the parent strains, which do not have nsAARS/tRNA(CUA), should also be shown.

The leakage and gain of nsAA incorporation are shown to a high degree of detail using the very sensitive NanoLuciferase reporter in Supp. Fig. 1G, and to a lesser degree using mNeongreenM10S in Supp. Fig. 1C. We feel that adding this information to figure 1 would make it significantly more difficult for the casual reader to understand.

page 7, line 8-9 “synthetase expression alone resulted in cellular retention of fluorescent nsAA

(Supp. Fig. 4A).”

The cognate tRNA(CUA) may be co-expressed.

We have corrected this section to say that synthetase & tRNA co-expression was sufficient for retention of fluorescent nsAA.

Page 7, line 12-13 “off-target nsAA incorporation”

Page 7, line 22; “misincorporation”

These terms should be defined.

We have removed these terms from the text, replacing them with “native protein incorporation”, which we define earlier in the text.

Page 9, Fig 2 and others

The identity of nsAA is indicated as their names (e.g., bipA, CouAA and pAzF). The compound number defined in Fig 1B (e.g., nsAA 1) is also used in other figures and text. The authors should use only either of them.

We have gone through the paper in detail to replace mentions of nsAA names with the appropriate numbers.

Page 10, Line 2 “2 incorporation was delayed”

nsAA 1 may be correct.

This correction has been made.

Page 10, line 3 Fig. 2D

Figure 2C may be correct.

This correction has been made.

Page 10, line 4; Fig. 2E

Fig. 2D may be correct.

This correction has been made.

Page 10, line 5; “1 could be incorporated during sporulation”

nsAA 2 may be correct.

This correction has been made.

Page 14, Figure 4A

The promoter driving TAG-mNeon should be indicated.

We have indicated this in the figure caption.

Page 14, Figure F

Z-ring cannot be seen at 3-10 μ M nsAA 2 in Figure 4C.

Under these conditions Z-rings become more sporadic and decondensed, appearing faint in these images. However, computational analysis of many cells did find sporadic Z-

rings, allowing us to find n>400 examples for each of these conditions, despite their rarity in any given image.

Page 17, line 13-14; “Combining transcriptional and translational titration allowed us to modulate FtsZ’s dynamics and therefore *B. subtilis* cell division.”

The legend of Fig.4 indicated that the concentration of IPTG was constant (100 μ M), suggesting that the authors did not use the combination of transcriptional and translational titration in this experiment.

We apologize for this oversight – the data included in the paper did only use translational titration instead of the combination as was originally planned. We have corrected this statement to accurately reflect our experiments.

Supp Fig. 3

The authors should clarify the following points:

(1) Was the tRNA(CUA) co-expressed?

(2) Was CouAA supplied in all experiments?

(3) The upper panel may be CouRS[+tRNA(CUA)] without TAG-mNeon. Otherwise, what is the “background fluorescence” (page 7, line 7)?

We have expanded the legend of this figure to clarify these questions. Briefly, 1 – the tRNA(CUA) was co-expressed with couRS in the bottom figure. 2 – yes, CouAA was supplied in all experiments. 3 – The upper panel only contains TAG-mNeon, the background fluorescence is likely due to some level of CouAA sticking to cells after washing. However, the bottom panel shows that CouAA fluorescence is increased dramatically when the AARS/tRNA are expressed.

Supp Fig 4A

Was the tRNA(CUA) co-expressed?

What is “TAG” (TAG-mNeon)?

Yes to both. We have corrected the figure and caption to clarify this point.

Supp Fig 5F

The OD-fluorescence curve for UAC+bipA, in which the normalized fluorescence was saturated at a lower OD, is largely different from others. The absolute value of bipA intake may be not enhanced.

This media, lacking organic nitrogen, is inhospitable to *B. subtilis*, which results in slower growth and later stationary according to Supp. Fig. 6F. While saturation of the positive control occurred at lower OD, nsAA incorporation became significant less than halfway to the OD where fluorescent saturation occurred. This is a significant difference to Supp. Fig 5A, where nsAA incorporation became significant much later, after growth had slowed.

//

Reviewer #2 (Remarks to the Author):

The manuscript by Stork et al. describes a new (expandable) toolbox for the incorporation of non-standard amino acids (nsAA) in *Bacillus subtilis*. *Bacillus subtilis* being a widely-used model species in a broad range of applications (fundamental and applied), this paper will be of interest to a large readership. Moreover, the authors bring attention on intriguing differences between *Bacillus subtilis* and *E. coli* (in which such tools have been extensively characterised), related to nsAA import and incorporation, in addition to their demonstration that the nsAA incorporation systems (nsAA + tRNA / aminoacyl-tRNA synthetase pairs) previously used in *E. coli* also work in *B. subtilis*. Overall this is a carefully conducted study including lots of interesting data for future work. I particularly appreciated that the authors raised awareness on (and extensively characterized) the limitations of their system. These limitations stem from the finding that nsAA incorporate at the proteome

level in a non-specific manner at UAG stop codons in *Bacillus*, implying that *Bacillus* must be more tolerant to proteome-wide amber stop codon suppression than *E. coli*. While no mechanism is investigated here, this idea will probably lead to interesting research paths and may lead to the discovery of major discrepancies between these two models regarding translation (termination) control. This non-specific incorporation hinders specific labelling strategies for now. I found their most interesting contribution being the possibility to drastically improve the dynamic range and tight control of protein levels in the cell by combining transcriptional control (via inducible promoters) and translational control (via controlled nsAA incorporation). The authors provide a proof of concept for this by modulating MciZ (an FtsZ inhibiting protein) levels, which has proven difficult so far due to the high sensitivity of FtsZ to even basal level of MciZ. In addition, they also incorporated UV-crosslinkable nsAA and performed *in vivo* cross-linking with particularly high efficiency, validating previous known interaction data. I only have minor comments.

We thank the reviewer for their kind comments, especially as it relates to characterizing the limitations of the system.

Fig 1.

- Panel D: was the signal normalised to the TAC-M10S-mNeogreen? If yes, adapt the Y axis label.

We have corrected the label.

Fig 2.

- A-B: it is unclear from the text or legend if the enrichment was performed on cells carrying the synthase alone as in Suppl Fig 4 or the corresponding tRNA too.

We have corrected the legend to explain that these experiments were performed on cells with both synthetase and tRNA.

Suppl Fig 1.

- Why are different concentration of nsAA used depending on the nsAA nature?

Though not well reported in the literature, the biochemical characteristics of the aminoacyl-tRNA synthetase regarding each substrate are known to vary, especially the affinity. Thus, sometimes a lower concentration than 1 mM can be used to obtain robust protein expression. We also look at the role of concentration later for one such nsAA,

pAzF, in modulating protein expression. In some cases, such as nsAA 1 & BiPyrA, higher concentrations of nsAA can precipitate or be toxic to cells at 1 mM and can therefore only be used at sub-millimolar concentrations. We now detail this rationale in the Supplementary text section.

- Panel A, it is unclear if the mNeonGreen construct in panel A is the UAG-mNeon or UAG-M10S-mNeon. Adapt if needed.

The incorrect panel A figure was previously used, displaying data from the original UAG-mNeon. We meant to display data using a corrected UAG-mNeonM10S, and have corrected this.

- Legend of panel A indicates normalisation to the max fluorescence from the experiment while the Y axis labels shows 'Fraction of UAC-mNeon fluor', similar to Fig 1D ('% of TAC-mNeongreen'). Please clarify what the normalisation is for Fig S1A and homogenize labels/legends across figures when appropriate.

This has also been corrected with replacement of the figure with corrected data. The figure now is normalized to UAC-mNeonM10S.

- Panels C-D: is there a reason for using abkRS in *E. coli* vs MaPyIRS in *Bacillus* for incorporating nsAA 5 (boc-K)?

AbkRS is an MbPyIRS, and is of limited activity in *B. subtilis*, as shown in Supplemental figure 1D. We have inserted additional discussion of this topic into the main text and figure legends of Figure 1 & Supp. Fig. 1.

Suppl Fig 2.

- Panel A, please indicate the expected size of mNeonGreen on the gel. There is a mistake in the MW marker labels. What is the small (~17 kDa) band found in all lanes except for bocK-containing mNeonGreen?

We have indicated the expected size of our mNeongreen reporter. With the elastin tag & a C-terminal His & FLAG tag it is the 32 kDa band. We do not know the identity of the 17kDa band, though we will note that both *B. subtilis* ferric uptake regulation protein & a superoxide dismutase-like protein are about 17 kDa and would be likely to co-purify with a cobalt-column his-tag purification.

- B; It is unclear where the elastin-like peptide is inserted in mNeonGreen.

We have expanded the sequence in Sup. Figure 2B to clarify this issue.

- Mass spectra and fragmentation patterns (C-H) are very small and impossible to read without zooming. Consider moving them in a spreadsheet.

We have reproduced the MS data in the raw data spreadsheet.

Reviewers' Comments:

Reviewer #1:

Remarks to the Author:

The revised version and the response from the author mostly alleviate the concerns which I originally had. Only I have a surrebuttal as described below, but those are all minor points.

[1]

(Reviewer 1, 1st round comment)

Page 9, Fig 2 and others

The identity of nsAA is indicated as their names (e.g., bipA, CouAA and pAzF). The compound number defined in Fig 1B (e.g., nsAA 1) is also used in other figures and text. The authors should use only either of them.

(Author's response)

We have gone through the paper in detail to replace mentions of nsAA names with the appropriate numbers.

(Reviewer 1, 2nd round comment)

In the following figures, the nsAA names are still used; Fig4A and most of the supplemental figures.

[2]

(Reviewer 1, 1st round comment)

Page 14, Figure F

Z-ring cannot be seen at 3-10 μ M nsAA 2 in Figure 4C.

(Author's response)

Under these conditions Z-rings become more sporadic and decondensed, appearing faint in these images. However, computational analysis of many cells did find sporadic Z-rings, allowing us to find $n > 400$ examples for each of these conditions, despite their rarity in any given image.

(Reviewer 1, 2nd round comment)

I recommend that the authors provide these explanations in the form of figures or text.

//

Reviewer #2:

Remarks to the Author:

The authors have addressed all my concerns in their rebuttal and in the revised manuscript.

REVIEWERS' COMMENTS

Reviewer #1 (Remarks to the Author):

The revised version and the response from the author mostly alleviate the concerns which I originally had. Only I have a surrebuttal as described below, but those are all minor points.

[1]

(Reviewer 1, 1st round comment)

Page 9, Fig 2 and others

The identity of nsAA is indicated as their names (e.g., bipA, CouAA and pAzF). The compound number defined in Fig 1B (e.g., nsAA 1) is also used in other figures and text. The authors should use only either of them.

(Author's response)

We have gone through the paper in detail to replace mentions of nsAA names with the appropriate numbers.

(Reviewer 1, 2nd round comment)

In the following figures, the nsAA names are still used; Fig4A and most of the supplemental figures.

We have corrected Figure 4A from pAzF to nsAA 2.

We have corrected text in the methods section that referred to pAzF, bipA & pBpA.

We have corrected supplemental figures and text that referred to various nsAAs by name, with the exception of Supp. Fig 1 E-F, which is the only place those nsAAs are referred to.

[2]

(Reviewer 1, 1st round comment)

Page 14, Figure F

Z-ring cannot be seen at 3-10 μ M nsAA 2 in Figure 4C.

(Author's response)

Under these conditions Z-rings become more sporadic and decondensed, appearing faint in these images. However, computational analysis of many cells did find sporadic Z-rings, allowing us to find $n > 400$ examples for each of these conditions, despite their rarity in any given image.

(Reviewer 1, 2nd round comment)

I recommend that the authors provide these explanations in the form of figures or text.

We have inserted this explanation into the caption for figure 4.

//

Reviewer #2 (Remarks to the Author):

The authors have addressed all my concerns in their rebuttal and in the revised manuscript.